# Apoptotic Effects of Drug Targeting Conjugates Containing Different GnRH Analogs on Colon Carcinoma Cells

**DOI:** 10.3390/ijms20184421

**Published:** 2019-09-08

**Authors:** Eszter Lajkó, Rózsa Hegedüs, Gábor Mező, László Kőhidai

**Affiliations:** 1Department Genetics, Cell- and Immunobiology, Semmelweis University, Nagyvárad tér 4, 1089 Budapest, Hungary; 2Research Group of Peptide Chemistry, Hungarian Academy of Sciences, Eötvös Loránd University, Pázmány Péter sétány 1/A, 1117 Budapest, Hungary; 3Eötvös Loránd University, Faculty of Science, Institute of Chemistry, Pázmány Péter sétány 1/A, 1117 Budapest, Hungary

**Keywords:** gonadotropin-releasing hormone, butyrate, conjugation, drug-targeting conjugates, impedimetry, daunorubicin, apoptosis, TP53, TNF, FASL

## Abstract

The wide range of cellular target reactions (e.g., antitumor) of gonadotropin-releasing hormone (GnRH) variants provides the possibility to develop multifunctional GnRH conjugates. The aim of our work was to compare the cytotoxic/apoptotic activity of different GnRH-based, daunorubicin (Dau)-linked conjugates with or without butyrated Lys in position 4 (^4^Lys(Bu)) at a molecular level in a human colorectal carcinoma cell line. Cell viability was measured by impedimetry, cellular uptake and apoptosis were studied by flow cytometry, and the expression of apoptosis-related genes was analyzed by qRT-PCR. The modification with ^4^Lys(Bu) resulted in an increased cytotoxic and apoptotic effects and cellular uptake of the GnRH-I and GnRH-III conjugates. Depending on the GnRH isoform and the presence of ^4^Lys(Bu), the conjugates could regulate the expression of several apoptosis-related genes, especially tumor necrosis factor (*TNF*), tumor protein p53 (*TP53*) and the members of growth-factor signaling. The stronger cytotoxicity of GnRH-I and GnRH-III conjugates containing ^4^Lys(Bu) was associated with a stronger inhibitory effect on the expression of growth-factor signaling elements in comparison with their ^4^Ser counterparts, in which the upregulation of *TP53* and caspases (e.g., *CASP9*) seemed to play a more important role. We were able to provide further evidence that targeting the GnRH receptor could serve as a successful therapeutic approach in colon cancer, and GnRH-III-[^4^Lys(Bu),^8^Lys(Dau=Aoa)] proved to be the best candidate for this purpose.

## 1. Introduction

Targeted tumor therapy represents a promising strategy to improve the selectivity and efficacy of chemotherapy by delivering a cytotoxic drug covalently linked to a targeting unit which is selective to a tumor’s overexpressed receptors. In many therapeutic approaches, hormone peptides have been proven to be a suitable carrier because their receptors were found to be overexpressed in many tumor types, while healthy cells lack these receptors [1]. One of these hormone peptides extensively studied and used for drug delivery is gonadotropin-releasing hormone (GnRH). Regarding targeting units, several native and synthetic GnRH analogs have been shown to efficiently affect only the tumor cells with a GnRH receptor (GnRH-R) and spare the healthy cells with no or a limited number of GnRH-Rs [2,3].

Originally, GnRH–GnRH-R interactions were identified as a central regulator of gonadal steroidogenesis and gametogenesis [4]. There are three main types of the decapeptide GnRH: GnRH-I, GnRH-II and GnRH-III. While the GnRH-I (mammalian GnRH-I: Glp-His-Trp-Ser-Tyr-Gly-Leu-Arg-Pro-Gly-NH_2_) and GnRH-II (chicken GnRH-II: Glp-His-Trp-Ser-His-Gly-Trp-Tyr-Pro-Gly-NH_2_) can be found in the human body [4,5], the GnRH-III (lamprey GnRH-III: Glp-His-Trp-Ser-His-Asp-Trp-Lys-Pro-Gly- NH_2_) is a non-human isoform; however, it can specifically bind to human GnRH-R [4,6]. Due to controversial findings of the existence of a functional GnRH-II receptor, it is highly suggested that the type-I GnRH-R (GnRH-IR) mediates the effects of the GnRH isoforms [4,7]. Their affinity and effects proved to be tissue/cell-dependent. At the pituitary level, GnRH-I binds to GnRH-R with high affinity and induces endocrine effects, while—depending on the tumor type—it can distinguish low or high-affinity binding sites, which mediate antiproliferative effects [4,8]. However, GnRH-III acts as a weak agonist to hypophyseal GnRH-IR and has a negligible endocrine effect, but in case of different tumors, it has the most effective antitumor activity compared to other isoforms [6,9,10]. Interestingly, GnRH-II exhibits the ability to inhibit the proliferation of both GnRH-IR positive and negative tumor cells. These findings imply the possibility of other types of binding partners on the tumor cell membrane [11,12].

GnRH peptides, as part of drug-delivery systems, have some valuable properties, such as (i) having a tumor growth inhibitory effect on their own (both agonists and antagonist possess this activity), and (ii) providing an easy method for modification and conjugation due to the well-studied structure–activity relationships [1]. One of the fundamental limitations of GnRH-based targeting is the relatively rapid proteolytic degradation of the peptide part [1,13]. Based on previous reports on structure–function relationships, the modification of GnRH-I and GnRH-II in position 6 with D-amino acid could increase their affinity to the GnRH-R [4] and their enzymatic stability as well as influencing the antitumor activity [12,14]. Concurrently with our own studies, this type of modification has been also utilized by many researchers for the design and synthesis of different drug-targeting conjugates by conjugating different chemotherapeutic agents (e.g., anthracyclines, methotrexate or cisplatin) to the side-chain of D-Lys of GnRH-I-[^6^D-Lys] and GnRH-II-[^6^D-Lys] [12,15,16,17]. In our previous work concerning the in vitro biological effects of different GnRH-I and GnRH-II-based conjugates, where daunorubicin (Dau) as an anticancer drug was attached directly or through an enzyme (cathepsin B) labile spacer (GFLG) to GnRH-I-[^6^D-Lys] or GnRH-II-[^6^D-Lys] via an oxime bound, we demonstrated that they have similar antiproliferative and apoptotic effects as well as receptor binding affinity [12].

In the case of GnRH-III, it was indicated that the modification of ^8^Lys, in order to form a conjugate or a dimer derivative, is allowed regarding its antiproliferative effect but results in a further decrease of its already depressed endocrine activity [10,18,19,20]. One of the most effective strategies to improve the antitumor activity and other biochemical properties of GnRH-III and its conjugate was to replace the Ser in position 4 with acylated Lys [13,21]. Based on the results of our systematic investigation of this kind of modification, GnRH-III-[^4^Lys(Bu),^8^Lys(Dau=Aoa)]—containing Dau in position 8 and an apoptosis-inducing butyrate as a “second drug” on the side chain of ^4^Lys—was proven to be the most potent conjugate due its enhanced antitumor activity, enzymatic stability and cellular uptake in different in vitro tumor models [21,22,23].

Various cytotoxic GnRH conjugates have been produced and investigated in our research group, whereby the oxime linkage was applied for attachment of Dau via an aminooxyacetyl (Aoa) moiety [3,12,19,21,23]. Due to the high chemical stability of the oxime bond conjugates, the premature release of the drug could be eliminated [19,24], which is supposed to be the main reason for the withdrawal of the ester bond-containing zoptarelin doxorubicin, a cytotoxic analog of GnRH-I (formerly known as AEZS-108 or AN-152) [25]. In these conjugates formed by oxime linkages, Dau could fulfil antineoplastic activity; however, this was not done by the intact Dau molecule but by the release of its amino acid-containing metabolites ((H-Lys(Dau=Aoa)-OH) or Dau=Aoa-Gly-OH, depending on the type of conjugates) [24]. Our previous studies also demonstrated that these Dau–GnRH hybrids are taken up by the GnRH-R positive target cells through receptor-mediated endocytosis, and the Dau-amino acid derivatives are cleaved off inside the cell [13,21,23] and bind to the DNA [24].

The unconjugated Dau can penetrate the plasma membrane by simple diffusion and trigger DNA damage by intercalation with the DNA and stabilization of the topoisomerase-II–DNA complex [26,27]. It is also evident that Dau can induce apoptosis independently from DNA interactions by influencing multiple signaling events, including the sphingomyelin–ceramide pathway, the generation of reactive oxygen species, the Fas-ligand (FASL)/FAS system, tumor suppressor gene TP53 and mitogen-activated/stress-activated kinases [27].

It has also been suggested that apoptosis is involved in the antitumor activity of different GnRH agonists and antagonists; however, the literature available on this issue is somewhat controversial. Depending on the cell type and GnRH analog, the extent of the apoptosis and its underlying pathways prove to be different. Goserelin, a GnRH-I agonist, has been demonstrated to increase the expression of the proapoptotic protein B-cell CLL/lymphoma 2 (BCL2)-associated X protein (BAX) in a TP53-dependent manner in a prostate cancer cell line [28], while in the case of ovarian cell lines, this analog could upregulate several members of tumor necrosis factor (TNF) and TNF-receptor superfamilies through the Forkhead Box O1–Phosphoinositide-3-Kinase–v-akt murine thymoma viral oncogene homolog (AKT) pathway [29]. The induction of anti-oncogenes was also shown to contribute to the apoptosis of multiple myeloma cells induced by the GnRH-I agonist [30]. The antagonist of both GnRH-I and GnRH-II was also reported to induce apoptosis in various tumor cell lines (e.g., prostate, adrenocortical, endometrial tumor cells). Different mechanisms, such as the activation of caspase 3/7, mitochondrial dysfunction (cytochrome C release), or the FASL/FAS-mediated pathway were involved in their apoptotic activity [31,32,33,34]. 

Depending on the target cells and the type of GnRH built in the anthracycline–GnRH conjugates, they could exhibit apoptotic activity to different extents. In the case of GnRH-I-[D-Lys^6^] or GnRH-II-[D-Lys^6^] with oxime-linked Dau, only a slight apoptotic activity could be determined on MCF-7 breast carcinoma cell line [12], while zoptarelin doxorubicin was proven to be more effective in inducing apoptosis in ovarian and endometrial carcinoma cell lines compared with the free drug [35]. Our recent results on melanoma cells indicated that in the case of the GnRH-III conjugate modified with butyrated ^4^Lys GnRH-III[^4^ Lys(Bu),^8^Lys(Dau=Aoa)]), its antitumor effect was rather attributed to apoptotic activity, in contrast to the GnRH-III[^8^Lys(Dau=Aoa)]-possessing ^4^Ser, where the cell cycle blocking effect (arrest in the G2/M phase) was shown to be more prominent [22]. Although an increasing number of studies have focused on the determination of the apoptosis induced by different cytotoxic GnRH compounds, there are limited data on the underlying molecular mechanism. For example, zoptarelin doxorubicin was proven to induce apoptosis by elevating levels of reactive oxygen species (ROS), Cyclin Dependent Kinase Inhibitor 1A (p21) and the rate of autophagy more significantly than the free doxorubicin in prostate cancer cells [26]. By investigating 87 apoptosis-related genes by a PCR array method, it was discovered that there were no major differences in the effects of the conjugate and the free drug. Several pro-apoptotic genes were increased by both compounds, but somewhat more strongly by zoptarelin doxorubicin. This conjugate could elevate the members of TNF and TNFR superfamilies (e.g., TNF receptor-associated factor 3 (*TRAF3*) and TNF receptor superfamily (TNFRSF)10B, 20, 25), the FASL/FAS pathway and different caspases (e.g., *CASP5, 8* and *10*) in PANC-1 pancreatic adenocarcinoma cells [36].

In most of the above-mentioned molecular biological studies on apoptosis, only GnRH-I analogs (e.g., goserelin) and conjugates (e.g., zoptarelin doxorubicin) were thoroughly investigated [26,29,30,36]. Nevertheless, the conjugates of GnRH-II and GnRH-III proved to have a comparable antitumor effect; moreover, they have been indicated to possess higher tumor selectivity and stability than that the GnRH-I-based conjugate [12,21,37,38]. The aim of our work was to compare the apoptotic activity of different GnRH-based, Dau-containing conjugates at the molecular level in a human colorectal carcinoma cell line. Based on these considerations, three GnRH analogs (GnRH-I-[^6^D-Lys], GnRH-II-[^6^D-Lys] and GnRH-III), previously proven to be effective in Dau-delivery, were selected to compare the apoptotic activity of their conjugates. Two sets of conjugates were synthesized by attaching Dau directly to the ^6^D-Lys or ^8^Lys (depending on the GnRH analog) via oxime linkage; one group with Ser in position 4 (GnRH-I-[^4^Ser,^6^D-Lys(Dau=Aoa)], GnRH-II-[^4^Ser,^6^D-Lys(Dau=Aoa)], GnRH-III[^4^Ser,^8^Lys(Dau=Aoa)]), and a second group in which this Ser was replaced with butyrated Lys (GnRH-I-[^4^Lys(Bu),^6^D-Lys(Dau=Aoa)], GnRH-II-[^4^Lys(Bu),^6^D-Lys(Dau=Aoa)], GnRH-III[^4^Lys(Bu),^8^Lys(Dau=Aoa)]) (Table 1). The idea behind examining these conjugates on colon carcinoma cells was that (i) GnRH-R is expressed on different colon cancer cells [39,40], (ii) different GnRH analog-containing conjugates could inhibit colorectal tumor growth both in vitro and in vivo [3,12,24,39,40,41], and (iii) modification with butyrated Lys has been already proven to be a successful strategy to improve the effectivity of conjugates in different types of GnRH-R-positive tumor cells (e.g., melanoma cells [22], colon and breast cancer cells [21,42,43]). In this paper, the synthesis and the analytical characterization of the above-mentioned GnRH conjugates with or without ^4^Lys(Bu) were described, while their antiproliferative and apoptotic activity were studied by impedimetry (xCELLigence SP System), flow cytometry and quantitative real-time RT-PCR in HT-29 human colon carcinoma cell line.

## 2. Results

### 2.1. Antitumor Effect of Dau–GnRH-[^4^Ser/^4^Lys(Bu)] Conjugates

In our previous studies, GnRH-R expression was verified on our HT-29 model cells by Western blot analysis [22,42]. In these studies, the short-term antitumor activity of Dau–GnRH conjugates with ^4^Ser as well as the enhanced cytostatic effect of the GnRH-III-based conjugate modified with ^4^Lys(Bu) were also determined on HT-29 cells by alamarBlue-assay [21,42]. However, which GnRH isoform could be the most suitable for targeting colon carcinoma cells and whether the incorporation of ^4^Lys(Bu) could be an effective way to increase the tumor growth inhibitory effect of these conjugates independently of the GnRH analog remained unclear. To compare the cytotoxic/antiproliferative effect of the conjugates with different native GnRH isoforms and to evaluate the effect of the substitution of butyrated Lys in position 4, a more sophisticated, impedance-based method (xCELLigence SP System) was used. The IC_50_–concentration required to decrease the cell viability by 50%—values showing the potency (Table 2) and cell viability (viab) showing the efficacy of the conjugates (Appendix A) were calculated from the results of the time-course study obtained after 24, 48 and 72 h of incubation time.

In case of the conjugates built on native GnRH conjugates, the II-[^4^Ser,^6^D-Lys(Dau)] proved to be the most potent, followed by the I-[^4^Ser,^6^D-Lys(Dau)] and III-[^4^Ser,^8^Lys(Dau)] (Table 2). Over the long term (after 72 h incubation), there was a significant difference (*p* < 0.014) in the cytotoxic efficacy of the GnRH-I and GnRH-II conjugates. However, I-[^4^Ser,^6^D-Lys(Dau)] showed the strongest antitumor effect (viab: 4.75%, Appendix A) at 10^−4^ M concentration, but according to the time-course study, the II-[^4^Ser,^6^D-Lys(Dau)] elicited a more immediate (Appendix A) cytotoxic effect (viab_24h_: 37.18% vs viab_24h_ for I-[^4^Ser,^6^D-Lys(Dau)]: 150.05%; Appendix A). III-[^4^Ser,^8^Lys(Dau)] had about a threefold weaker IC_50_ value (Table 2) and the maximal tumor growth inhibitory effect was manifested at 10^−4^ M concentration (viab_72h_: 19.05%) and only after 72 h incubation (Appendix A).

The substitution with ^4^Lys(Bu) proved to modify the cytotoxic effect of the conjugates depending on the type of GnRH analog. The most significant change was detected in case of the GnRH-III-based conjugates. As was expected, the replacement of ^4^Ser by ^4^Lys(Bu) led to a more than one order of magnitude smaller IC_50_ value (Table 2) and a stronger antitumor activity with an earlier onset (Appendix A, Appendix A). In the case of GnRH-I conjugates, this type of modification could cause only a slight increase in the potency (smaller IC_50_ value) after 72 h, but the onset of the cytotoxic activity took less time (48 h for I-[^4^Lys(Bu),^6^D-Lys(Dau)] vs. 72 h for I-[^4^Ser,^6^D-Lys(Dau)]. On the contrary, IC_50_ values of GnRH-II conjugate with ^4^Lys(Bu) (II-[^4^Lys(Bu),^6^D-Lys(Dau)]) were more than two times higher than those of II-[^4^Ser,^6^D-Lys(Dau)] after 48 and 72 h of incubation (Table 2).

The real-time data showed that all conjugates (except for III-[^4^Lys(Bu),^8^Lys(Dau)]) cause an initial (0–30 h) increase in the cell index values in 4 and 20 µM concentrations compared to the control, but over the long term (after ~40 h), the cell index values constantly decreased (Appendix A). This profile of the real-time curves could be caused by morphological changes induced by the conjugates. The lower dose of Dau-containing conjugates might cause cellular senescence—an irreversible growth arrest, which might evolve cell death (e.g., apoptosis) by increasing the concentration and/or incubation time [44,45]. Senescent cells are characterized by a large and flat cell morphology and consequently higher cell index values, while in the case of apoptosis, the cells round up and detach from the electrode surface leading to a decrease in cell index values [45,46].

### 2.2. Cellular Uptake of Dau–GnRH–[^4^Ser/^4^Lys(Bu)] Conjugates by HT-29 Cells

In order to investigate the internalization ability of the conjugates, HT-29 cells were incubated with the conjugates for 6 h, and a flow cytometric study was carried out. Only living cells were gated to evaluate the intracellular fluorescent intensity—expressed as GeoMean (geometric mean channel) value—of the incorporated Dau. The results of the internalization measurement corroborated well with the cytotoxic effects of the conjugates. The GnRH-I and GnRH-III conjugates with ^4^Lys(Bu) had an improved cellular uptake in comparison with the corresponding ^4^Ser derivatives, while II-[^4^Lys(Bu),^6^D-Lys(Dau)] showed reduced internalization over ^4^Ser counterparts (Figure 1). In the case of the Ser^4^-containing conjugates, the II-[^4^Ser,^6^D-Lys(Dau)] was taken up by HT-29 cells more effectively than the GnRH-I and GnRH-III conjugates. Furthermore, the cellular uptake of III-[^4^Lys(Bu),^8^Lys(Dau)] proved to be the highest compared to all the tested conjugates (Figure 1).

### 2.3. Apoptotic Effect of Dau–GnRH–[^4^Ser/^4^Lys(Bu)] Conjugates Detected by Flow Cytometry

The apoptotic cell death induced by 24 h of incubation with GnRH conjugates was measured by detecting the binding of fluorescein isothiocyanate (FITC)-conjugated annexin V.

In general, the conjugates had a minor or no apoptotic effect. In the case of conjugates with Ser^4^, only the GnRH-II conjugate could elicit a slight, but significant, apoptotic effect, and the incorporation of ^4^Lys(Bu) diminished this activity (Figure 2). Among the tested conjugates, III-[^4^Lys(Bu),^8^Lys(Dau)] had the maximal apoptotic effect, while there was no significant difference between III-[^4^Ser,^8^Lys(Dau)] and the control (Figure 2).

### 2.4. Effect of Dau–GnRH–[^4^Ser/^4^Lys(Bu)] Conjugates on the Expression of Human Apoptosis-Related Genes

To investigate the molecular background of the HT-29 cell death induced by GnRH conjugates, a human apoptosis gene PCR array (RealTime ready custom panel, Roche Applied Science, Mannheim, Germany) containing 23 apoptosis-related genes was used. During the evaluation, genes with an expression fold change ≥2 were taken into consideration.

The expression of genes in HT-29 cells treated with the different GnRH conjugates with Ser in position 4 is shown in Figure 3. GnRH-I and GnRH-III conjugates with ^4^Ser increased the expression of *TP53* after 24 h (3.13 and 2.65 fold, respectively) and 48 h (5.74 and 2.53 fold, respectively) incubation (Figure 3A,C), while II-[^4^Ser,^6^D-Lys(Dau)] did not affect the *TP53* level (Figure 3B). All of the conjugates containing ^4^Ser could increase the expression of genes involved in the intrinsic pro-apoptotic pathway, but with different activity. For example, in the case of BCL2-associated agonist of cell death (*BAD*), the order of activity after 48 h was the following: I-[^4^Ser,^6^D-Lys(Dau)] (9.39 fold) > II-[^4^Ser,^6^D-Lys(Dau)] (6.82 fold) > III-[^4^Ser,^8^Lys(Dau)] (3.35 fold) (Figure 3). The *BCL2* expression was completely abolished after 48 h incubation with I-[^4^Ser,^6^D-Lys(Dau)] and II-[^4^Ser,^6^D-Lys(Dau)] (Figure 3A,B). The ^4^Ser conjugates caused the most remarkable increase in the case of *TNF* expression, especially after 24 h (I-[^4^Ser,^6^D-Lys(Dau)]: 40.57-fold, II-[^4^Ser,^6^D-Lys(Dau)]: 18.33-fold, III-[^4^Ser,^8^Lys(Dau)]: 50.45) (Figure 3). This stimulatory effect on *TNF* became smaller after 48 h of treatment. In parallel, all of the ^4^Ser conjugates also upregulated *FASL* (I-[^4^Ser,^6^D-Lys(Dau)]: 17.07-fold, II-[^4^Ser,^6^D-Lys(Dau)]: 7.66-fold, III-[^4^Ser,^8^Lys(Dau)]: 7.43-fold) in a time-dependent manner. In the expression of *FAS* and Fas (TNFRSF6)-associated via death domain (*FADD*), minor positive (I-[^4^Ser,^6^D-Lys(Dau)] or negative (II-[^4^Ser,^6^D-Lys(Dau)]) changes were detected (Figure 3). Basically, only the I-[^4^Ser,^6^D-Lys(Dau)] could influence (increase) the expression of *CASP9, 7* and *3* (2.09, 2.23 and 2.68 fold, respectively) (Figure 3A). The level of *CASP8* was decreased after 48 h by all conjugates with ^4^Ser (Figure 3), but this effect was more prominent in the II-[^4^Ser,^6^D-Lys(Dau)]-treated group (0.15-fold) (Figure 3B). All of the ^4^Ser conjugates upregulated the expression of signal transducer and activator of transcription 1 (*STAT1*) with comparable activity (I-[^4^Ser,^6^D-Lys(Dau)]: 4.71-fold, II-[^4^Ser,^6^D-Lys(Dau)]: 2.67-fold, III-[^4^Ser,^8^Lys(Dau)]: 5.55-fold) (Figure 3). The expression of other tested genes involved in the growth factor signaling pathway was reduced by II-[^4^Ser,^6^D-Lys(Dau)], and this activity became more pronounced after 48 h incubation (v-rel reticuloendotheliosis viral oncogene homolog A (*RELA*): 0.09-fold, nuclear factor of kappa light polypeptide gene enhancer in B-cells 1 (*NFKB1*): 0.17-fold, phosphatase and tensin homolog (*PTEN*): 0.18-fold, v-akt murine thymoma viral oncogene homolog 1 (*AKT1*): 0.03-fold) (Figure 3B). The ^4^Ser containing GnRH-I and GnRH-III conjugates had milder activity for this group of the genes: they could decrease the expression only of *PTEN* (0.24 and 0.28, respectively) and *AKT1* (0.11 and 0.18, respectively) (Figure 3A,C). The expression of anti-apoptotic genes was elevated by the conjugates with ^4^Ser. In the case of II-[^4^Ser,^6^D-Lys(Dau)], both heat shock protein 90 kDa beta (Grp94) member 1 *(HSP90B1*) (2.08-fold) and suppressor of cytokine signaling 2 (*SOCS2*) (10.99-fold) were affected (Figure 3B), while I-[^4^Ser,^6^D-Lys(Dau)] and III-[^4^Ser,^8^Lys(Dau)] could increase the expression only of *SOCS2* (11.56 and 4.29, respectively) in a long-term manner (Figure 3A,C).

The replacement of ^4^Ser by ^4^Lys(Bu) led to significant changes in the expression of apoptosis-related genes, as shown in Figure 4 (24 h treatment) and Appendix A (48 h treatment). When the cells were treated with ^4^Lys(Bu) containing GnRH-I or GnRH-III, the expression of the *TP53* was decreased to the control level or below in comparison with ^4^Ser derivatives (Figure 4A,C and Appendix A). On the contrary, II-[^4^Lys(Bu),^6^D-Lys(Dau)] upregulated the *TP53* (2.32 fold after 24 h) compared to the control and II-[^4^Ser,^6^D-Lys(Dau)] (Figure 4B). Similar to ^4^Ser conjugates, all of the conjugates with ^4^Lys(Bu) increased the level of *BAD* (I-[(^4^Lys(Bu),^6^D-Lys(Dau)]-I: 4.87-fold, II-[^4^Lys(Bu),^6^D-Lys(Dau)]: 4.28-fold and III-[^4^Lys(Bu),^8^Lys(Dau)]: 2.55-fold) and *BAX* (I-[(^4^Lys(Bu),^6^D-Lys(Dau)]: 2.25-fold, II-[^4^Lys(Bu),^6^D-Lys(Dau)]: 3.71-fold and III-[^4^Lys(Bu),^8^Lys(Dau)]: 2.53-fold) as well as abolishing *BCL2* expression (Figure 4 and Appendix A). The most remarkable difference between the ^4^Ser and ^4^Lys(Bu) conjugates was detected in the expression of *TNF*. Due to the 24 h treatment with the ^4^Lys(Bu) conjugates, the expression of *TNF* was not detectable (Figure 4). A slight increase was shown in the *TNF* level (2.87-fold) after the cells were incubated with III-[^4^Lys(Bu),^8^Lys(Dau)] for 48 h (Appendix A). The cells treated with ^4^Lys(Bu) derivatives could not only overexpress the *FASL* I-[(^4^Lys(Bu),^6^D-Lys(Dau)]: 6.36-fold, II-[(^4^Lys(Bu),^6^D-Lys(Dau)]: 9.00-fold, III-[^4^Lys(Bu),^8^Lys(Dau)]-III: 5.40-fold) (as in case of the ^4^Ser conjugates) (Figure 4 and Appendix A) but also the *FADD* (I-[(^4^Lys(Bu),^6^D-Lys(Dau)]: 4.3-fold, II-[(^4^Lys(Bu),^6^D-Lys(Dau)]: 6.22-fold and III-[^4^Lys(Bu),^8^Lys(Dau)]: 2.16-fold), especially after 48 h (Appendix A). Expression of *FAS* varied in different directions: decreases were seen in cells treated with I-[^4^Lys(Bu),^6^D-Lys(Dau)] (0.32-fold) and III-[^4^Lys(Bu),^8^Lys(Dau)] (0.07-fold) after 48 h (Appendix A), while increases were observed in the case of II-[^4^Lys(Bu),^6^D-Lys(Dau)] (3.09-fold) (Appendix A). The gene expressions of the effector proteins and members of the growth-factor signaling pathway were greatly reduced by III-[^4^Lys(Bu),^8^Lys(Dau)] (Figure 4C and Appendix A). In contrast with the ^4^Ser counterparts, GnRH-I and GnRH-II conjugates with ^4^Lys(Bu) caused an increase in the expression of *CASP8* (I-[(^4^Lys(Bu),^6^D-Lys(Dau)]: 2.99-fold, II-[(^4^Lys(Bu),^6^D-Lys(Dau)]: 3.88-fold) and *CASP7* (I-[(^4^Lys(Bu),^6^D-Lys(Dau)]: 2.99-fold, II-[^4^Lys(Bu),^6^D-Lys(Dau)]: 2.59-fold) after 24 h of incubation, while the rest of the caspase genes were not affected (Figure 4A,B). 

I-[^4^Lys(Bu),^6^D-Lys(Dau)] and III-[^4^Lys(Bu),^8^Lys(Dau)] downregulated *HMGB1* expression by a ca. 3-fold change (Figure 4A,C and Appendix A), while among the ^4^Ser conjugates, only the III-[^4^Ser,^8^Lys(Dau)] had this activity after 24h (Figure 4C). The expression of *STAT1* was changed in parallel with the *TP53* (Figure 4 and Appendix A); both genes were increased in the II-[^4^Lys(Bu),^6^D-Lys(Dau)]-treated group (3.00 and 2.32, respectively) (Figure 4B), or decreased in the case of III-[^4^Lys(Bu),^8^Lys(Dau)] (0.01 and 0.36, respectively) (Figure 4C) after 24 h treatment. Except for *RELA* and *NFKB1* in the II-[^4^Lys(Bu),^6^D-Lys(Dau)]-treated group, all of the genes involved in the growth-factor signaling pathway were reduced by the ^4^Lys(Bu) conjugates (Figure 4 and Appendix A). The effects of ^4^Lys(Bu) conjugates on *SOCS2* expression were similar to that of the ^4^Ser derivatives: at least a 4–10-fold increase was detected after 48 h (Appendix A). The expression of anti-apoptotic *HSP90B1* was variable in the cells treated with the three conjugates with ^4^Lys(Bu) (I-[^4^Lys(Bu),^6^D-Lys(Dau)]: 2.00, II-[^4^Lys(Bu),^6^D-Lys(Dau)]: no change, III-[^4^Lys(Bu),^8^Lys(Dau)]: 0.44-fold) (Figure 4).

## 3. Discussion

The development of peptide-based targeted agents is a welcome addition to traditional chemotherapy and even biological therapy for patients with advanced colorectal cancer. Receptors for various peptide hormones have been demonstrated in colorectal carcinomas, and these receptors could be utilized for both therapeutic and diagnostic purposes [47,48]. A couple of publications have reported the expression of GnRH-R in different colon cancer cell lines [39,49,50]. Furthermore, different GnRH-R agonists and antagonists were successfully utilized in drug-targeting conjugates to inhibit their proliferation [12,13,21,39,51,52]. One of the first conjugates targeting different GnRH-R-positive tumor cells (e.g., ovarian, endometrial or colon carcinomas) was designed in Schally’s lab by attaching doxorubicin to the GnRH-I-[^6^D-Lys] superagonist via an ester bond [17,39] and approved as zoptarelin doxorubicin [53]. Nevertheless, the primary endpoint (median overall survival) was not improved with this conjugate in a phase III endometrial cancer trial compared to the free drug [54]. It was assumed that the main reason for this failure was the poor stability of the ester bond in the human serum and consequently the premature release of doxorubicin [17]. To overcome this weakness and to improve cardiotoxic and/or endocrine side-effect-free antitumor activity, more stable drug-linkers (e.g., oxime bond) and more tumor-selective GnRH analogs could be advantageous. The GnRH analogs are widely known for their antitumor activity, which represents a valuable benefit for their use in targeted therapy [1]. It was shown that GnRH-II and GnRH-III were found to be more potent than GnRH-I in inhibiting the proliferation of breast cancer cells [9], for example, and the GnRH-III has an insignificant gonadotropin-releasing effect compared to GnRH-I and GnRH-II [10]. Previous structure–activity relationship studies have reported that the replacement of ^6^Gly in GnRH-I and GnRH-II [12] and the elimination of the basic character of ^8^Lys in GnRH-III were allowed with respect to their antiproliferative effect [10]. The same results were demonstrated for position 4 [10,13]. When ^4^Ser was exchanged by acylated (especially butyrated) ^4^Lys in the conjugates of GnRH-III and Dau, their antitumor activity was significantly improved [13,21].

The present paper reports the first study into the comparison of the cytotoxic and apoptotic effects of conjugates containing different GnRH isoforms with native Ser in position 4 or their derivatives substituted with butyrated ^4^Lys. All of the six conjugates carried an oxime-linked Dau in the side chain of ^6^D-Lys or ^8^Lys depending on the GnRH derivatives. The aim of our present work was to find which GnRH isoform is optimal for targeting colon carcinoma cells and whether the exchange of ^4^Ser by ^4^Lys(Bu) could be a universal strategy to improve the efficacy of any kind of GnRH conjugates.

After the synthesis and the analytical characterization of the conjugates, the impedimetric measurement revealed that the GnRH-I-[^4^Ser] and GnRH-II-[^4^Ser] conjugates had comparable tumor growth inhibitory effects in HT-29 cells, which confirms our previous findings [12]. Although the GnRH-III conjugate with ^4^Ser proved to be the least effective, its ^4^Lys(Bu) counterparts showed the strongest cytotoxic activity. In the case of GnRH-III conjugates, the beneficial effect of this modification with ^4^Lys(Bu) was well-established in our previous studies [21,43]. Thus, this pair of GnRH-III conjugates were considered as reference conjugates in our present study. In the case of the GnRH-I conjugates, the modification with ^4^Lys(Bu) could also increase the antitumor activity, while a decreased cytotoxicity was determined for II-[^4^Lys(Bu),^6^D-Lys(Dau)]. Similar differences in the apoptotic activity of the conjugates with ^4^Ser were observed, and the incorporation of ^4^Lys(Bu) also resulted in a similar change in their apoptotic effect, as seen by the cytotoxicity assay. A good correlation was found between the internalization ability and the cytotoxic and apoptotic effects of the conjugates. The order of the antitumor activity and the internalization rate was more or less the same. It seemed that the cellular uptake of the conjugates basically determines their antitumor activity in HT-29 cells. According to our previous findings, the incorporation of ^4^Lys(Bu) into a Dau-containing GnRH-III conjugate led to increased receptor binding affinity [23,42], which might explain the enhanced cellular uptake and consequently the stronger antitumor activity. The similar change in the effects of GnRH-I and GnRH-III as a result of the incorporation of ^4^Lys(Bu) indicated that this modification could also enhance the receptor affinity of GnRH-I, while in the case of GnRH-II, the native Ser in position 4 appeared to be more important in the receptor binding and/or receptor activation.

By detecting the expression of 23 apoptosis-related genes simultaneously with a PCR array, we were able to discover some similarities between the effects of I-[^4^Ser,^6^D-Lys(Dau)] and III-[^4^Ser,^8^Lys(Dau)] conjugates. For example, both of them increased the mRNA expression of *TP53*, some elements of the intrinsic apoptotic pathway (e.g., *BAD*, *BAK1*), and, most significantly, the *TNF*-level. In contrast, II-[^4^Ser,^6^D-Lys(Dau)] was found to mediate its antitumor effects by interfering with growth-factor signaling and regulating the intrinsic apoptotic pathway (e.g., *BAX* and *BCL2*), but independently of *TP53*. Although the expression of extrinsic pro-apoptotic factors (e.g., *TNF* and *FASL*) was increased all ^4^Ser conjugates, the GnRH-II version had a somewhat lesser effect on them. In the case of the caspase dependency, another major difference between the ^4^Ser conjugates was observed. While the effect of II-[^4^Ser,^6^D-Lys(Dau)] proved to be caspase-independent, the GnRH-I and GnRH-III conjugates with ^4^Ser upregulated the caspase expression. However, III-[^4^Ser,^8^Lys(Dau)] had a less significant effect (increase only in case of *CASP9*) than I-[^4^Ser,^6^D-Lys(Dau)], which could increase the expression of *CASP3, 7* and *9*. Our results match well with previous findings on different model cells. Goserelin, a GnRH-I agonist, was shown to increase the BAX expression in a TP53-dependent manner in prostate cancer cells [28]. In the case of ovarian carcinoma and multiple myeloma cells, the enhancement of TP53 expression was also demonstrated against the background of the antitumor effect of different GnRH-I agonists and antagonists [30,55]. Zhang and his coworkers have recently reported that goserelin induced apoptosis in epithelial ovarian cancer cells by partly upregulating factors of the TNF and TNF-receptor superfamilies [29]. Several studies have demonstrated that treatment with a GnRH-I analog could induce FASL expression in different reproductive cancer cells [55,56]. Regarding GnRH-II, recent studies also found that GnRH-II and its analogs elicited an antitumor effect in ovarian and endometrial cancer cell lines, for example, by counteracting growth factor-induced mitogenic signaling [55,56]. Only a couple of studies are available in the literature which deal with the apoptotic inducer effect [12,22,26], or with the molecular biological background of the cytotoxic GnRH conjugates [36,57]. Szepesházi and his co-workers analyzed the effect of zoptarelin doxorubicin in different pancreas carcinoma cell lines. There was a relatively good agreement between their results on PANC-1 cells, for example, and our results on HT-29, since the zoptarelin doxorubicin was also found to increase the expression of *TP53*, *FASL* and other members of the FAS signaling pathway, pro-apoptotic members of intrinsic pathways and different caspases [36].

Compared to the ^4^Ser GnRH-I, -II and –III conjugates, their ^4^Lys(Bu) counterparts had significantly different effects on the expression of apoptosis-related factors, as well. Nevertheless, it is worth mentioning that, independently of this modification, all of the investigated conjugates could increase the expression of *FASL* and the mitochondrial pro-apoptotic factors (e.g., *BAD* and *BAX*), which indicates the general importance of the FAS-dependent pathway and the mitochondrial apoptotic pathway in the antitumor effect of different GnRH conjugates. However, further studies are needed to establish the causative linkage between these apoptosis markers and the antiproliferative/cytotoxic effect of GnRH conjugates. One of the most significant changes due to ^4^Lys(Bu) was recognized in the expression of *TP53* and *TNF*. Compared to the control, the presence of ^4^Lys(Bu) resulted in a remarkable downregulation (e.g., *TNF* in case of all three ^4^Lys(Bu) conjugates) or in a neutral effect (e.g., *TP53* in case of I-[^4^Lys(Bu),^6^D-Lys(Dau)]) in their expression, while the *TP53* and *TNF* seemed to play an important role in the apoptotic effect of the ^4^Ser conjugates. In the case of the GnRH-II conjugate pair, our PCR results also indicated that the sequence modification with ^4^Lys(Bu) caused an inverse change in the expression of apoptosis-related proteins compared to the sets of GnRH-I and GnRH-III conjugates. The antitumor effect of the I-[^4^Lys(Bu),^6^D-Lys(Dau)] and III-[^4^Lys(Bu),^8^Lys(Dau)] appeared to be rather more caspase-independent than in the case of their ^4^Ser derivatives. However, the ^4^Lys(Bu)-containing GnRH-II conjugate was characterized by caspase dependency. The best example of this kind of antagonistic change was seen in the case of the markers of growth-factor signaling. The expressions of *RELA*, *NFKB1*, and *AKT1,* for example, were significantly downregulated in the cells treated with GnRH-I and GnRH-III conjugates containing ^4^Lys(Bu), while after the treatment with II-[^4^Lys(Bu),^6^D-Lys(Dau)], their expression was upregulated compared to the ^4^Ser conjugates. Based on our results, we could demonstrate that the stronger cytotoxic activity of I-[^4^Lys(Bu),^6^D-Lys(Dau)], III-[^4^Lys(Bu),^8^Lys(Dau)] and II-[^4^Ser,^6^D-Lys(Dau)] was associated with a stronger inhibitory effect on the expression of elements of growth-factor signaling. In the case of their counterparts, the upregulation of the expression *TP53*, *TNF* and caspases (e.g *CASP9*) probably had a more important role in the antitumor activity of I-[^4^Ser^6^D-Lys(Dau)], III-[^4^Ser,^8^Lys(Dau)] and II-[^4^Lys(Bu),^6^D-Lys(Dau)]. Formerly, we investigated the effect of a similar GnRH-III derivative conjugate (modified in position 4 with acetylated Lys (^4^Lys(Ac)) on the protein expression profile of HT-29 human colon cancer cells by mass spectrometry-based proteomics [57]. Although there was no overlap between the results of our former and present study, some proteins exerting anti-apoptotic activity (e.g., heat shock 70 kDa protein 1A/1B [58] or protein disulfide isomerase [59]) were found to have a decreased effect after treatment with GnRH-III conjugate-containing ^4^Lys(Ac) [57]. It should be noted that there were some differences between these studies; for example, the treatment conditions (the type of conjugate, incubation time, etc.), the methods and the stages of expression (gene expression vs. protein level) being investigated were different. Despite these differences, the results of our previous and present study clearly indicate that GnRH conjugates exert their cytotoxic action by modulating/interfering with multiple intracellular processes including apoptosis and cell growth.

The expression of markers involved in the different (e.g., intrinsic, or extrinsic) apoptotic signaling pathways were characterized by different time dependencies. In general, the upregulation of *TP53*, *TNF* and caspase expression proved to be an early change. Most of the conjugates could increase the expression of *FASL* and members of the pro-apoptotic mitochondrial pathway after 24 h of incubation, and this effect persisted or became more significant after a long time. On the contrary, the increase in the anti-apoptotic protein expression was observed only in a long-term manner (after 48 h). I-[^4^Lys(Bu),^6^D-Lys(Dau)] and III-[^4^Lys(Bu),^8^Lys(Dau)] could inhibit the expression of growth-factor signaling proteins already after a 24 h incubation, and this effect intensified by 48 h. It was more typical for GnRH-I and GnRH-III conjugates with ^4^Ser that they inhibited these markers only in a long-term manner.

In summary, we demonstrated that the modification with Lys(Bu) in position 4 resulted in an increase in the cytotoxic and apoptotic effect and the internalization ability of GnRH-I and GnRH-III conjugates containing an oxime-linked Dau. Although the conjugates had a minor apoptotic effect, they could regulate the expression of several apoptosis-related factors, and this activity proved to be sensitive to the GnRH isoforms and the presence of the ^4^Lys(Bu), especially in the case of *TNF*, *TP53* and members of the growth-factor signaling pathway. By detecting the expression of 23 apoptosis-related genes, we could find further evidence that the GnRH-I and GnRH-III conjugates acted in a more or less similar way. Our comprehensive PCR results show that the stronger cytotoxic activity of I-[^4^Lys(Bu),^6^D-Lys(Dau)], III-[^4^Lys(Bu),^8^Lys(Dau)] and II-[^4^Ser,^6^D-Lys(Dau)] was associated with a stronger and a more immediate inhibitory effect on the expression of the elements of growth-factor signaling compared to their counterparts, where the upregulation of the expression of *TP53*, *TNF* and caspases (e.g., *CASP9*) probably had a more important role. Our results also suggest the significance of ^4^Lys(Bu) in the receptor binding or activation of GnRH-I and GnRH-III conjugates, while in the case of GnRH-II conjugates, the native Ser in position 4 appeared to be more important. We were able to provide further evidence that targeting the GnRH-R could serve as a successful therapeutic approach in colon cancer, and III-[^4^Lys(Bu),^8^Lys(Dau)] was proven to be the best candidate for this purpose.

## 4. Materials and Methods

### 4.1. Material

All amino acid derivatives and Rink-Amide MBHA resin were purchased from Iris Biotech GmBH (Marktredwitz, Germany). Boc-aminooxyacetic acid (Boc-Aoa-OH), coupling agents and cleavage reagents (1-hydroxybenzotriazole hydrate (HOBt), *N*,*N*′-diisopropylcarbodiimide (DIC), triisopropylsilane (TIS), piperidine, 1,8-diazabicyclo[5.4.0]undec-7-ene (DBU), trifluoroacetic acid (TFA)), diisopropylethylamine (DIPEA), *n*-butyric anhydride, and acetonitrile (MeCN) for HPLC were obtained from Sigma-Aldrich Kft. (Budapest, Hungary). Daunorubicin hydrochloride was a gift from IVAX (Budapest, Hungary). Dimethylformamide (DMF), dichloromethane (DCM) and diethyl ether were purchased from Molar Chemicals Kft (Budapest, Hungary). All reagents and solvents were of analytical grade or highest available purity.

### 4.2. Synthesis of GnRH-Based Conjugates

GnRH derivatives were built up by solid-phase peptide synthesis (SPPS) on Rink-Amide MBHA resin using Fmoc/tBu strategy as described previously [21]. Briefly, Fmoc-Lys(Mtt)-OH was incorporated to position 8 for GnRH-III or Fmoc-D-Lys(Mtt)-OH in position 6 for GnRH-I and GnRH-II derivatives, and Fmoc-Lys(ivDde)-OH in position 4 of the peptide sequence in the case of butyrylated variants. Otherwise, standard amino acid derivatives were applied. After the attachment of the last amino acid (pyroglutamic acid: Glp), first, the orthogonal ivDde-protecting group was removed with 4% hydrazine in DMF followed by the acylation of the free ε-amino group of ^4^Lys with three equivalent butyric anhydride. In the next step, the Mtt-protecting group was cleaved selectively with 2% TFA and 2% TIS in DCM, and after neutralization (10% DIPEA in DCM), Boc-Aoa-OH was coupled for 2 h using DIC and HOBt coupling reagents (three equivalent each to the amino group). Finally, the functionalized peptide derivatives were detached from the resin using a mixture of 95% TFA, 2.5% TIS and 2.5% water (*v*/*v*/*v*). After purification of the crude product, Dau was linked by chemoselective ligation to the aminooxiacetylated peptides via oxime bond formation in 0.2 M ammonium acetate buffer (pH 5.0) at a peptide concentration of 10 mg/mL. The conjugates were purified by high-performance liquid chromatography (HPLC) followed by lyophilization. The purity and identity of the products were analyzed by analytical HPLC and electrospray ionization–mass spectrometry (ESI-MS).

### 4.3. Analytical Characterization of Conjugates

#### 4.3.1. RP-HPLC

A KNAUER 2501 HPLC system (H. Knauer, Bad Homburg, Germany) using a semipreparative Phenomenex Luna C18 column (250 mm × 10 mm) with 10 mm silica (100 Å pore size) (Torrance, CA, USA) was used. A linear gradient elution (0 min 20% B; 5 min 20% B; 50 min 100% B) with eluent A (0.1% TFA in water) and eluent B (0.1% TFA in MeCN–H_2_O (80:20, *v*/*v*)) was used at a flow rate of 4 mL/min. Peaks were detected at λ = 280 nm.

Analytical RP-HPLC was performed on a KNAUER 2501 HPLC system using a Phenomenex Luna C18 column (250 mm × 4.6 mm) with 5 mm silica (100 Å pore size). Linear gradient elution (0 min 0% B; 5 min 0% B; 50 min 90% B or 0 min 0% B; 2 min 0% B; 25 min 100% B) at a flow rate of 1 mL/min with the eluents described above. Peaks were detected at λ = 220 nm.

#### 4.3.2. ESI-MS

Electrospray ionization (ESI)-mass spectrometric analyses were carried out on an Esquire 3000þ ion trap mass spectrometer (Bruker Daltonics, Bremen, Germany). Spectra were acquired at 50–2500 *m*/*z*.

### 4.4. Cell Culture

The HT-29 human colon adenocarcinoma cell line, obtained from the European Collection of Authenticated Cell Cultures (ECACC, Salisbury, UK), was used for our model cells. RPMI medium supplemented with 10% heat-inactivated fetal bovine serum (FBS, Gibco^®^/Invitrogen Corporation, New York, NY, USA), l-glutamine (2 mmol/L) (Lonza, Basel, Switzerland) and 100 µg/mL penicillin/streptomycin (Gibco^®^/Invitrogen Corporation, New York, NY, USA) was used for cell culturing. Cells were grown at 37 °C in a humidified 95% air/5% CO_2_ atmosphere.

For cell counting, the impedance-based CASY TT Cell Counter and Analyser (Innovatis AG, Roche Applied Science, Mannheim, Germany) was applied to differentiate between viable and dead cells as well as to determine overall sample viability. The principles and the procedure of the measurements were described in our previous paper [60].

### 4.5. Cell Viability Assay

The effects of different GnRH conjugates on HT-29 cell growth were measured by the xCELLigence SP System (ACEA Biosciences, San Diego, CA, USA), which allows cell viability to be detected in real-time via impedance readouts. The experimental setup and the procedure were almost identical to the one described in more detail in our previous works [22,61]. The continuous monitoring of cell viability provides the opportunity to determine the difference in the time-dependent cellular response for the various GnRH conjugates and to select the optimal time points to conduct more specific end-point assays (e.g., apoptosis assays) for further investigating the mechanism of antitumor activity. The impedance change of the electrode on the bottom of a so-called E-plate is displayed in real-time as the cell index (CI). CI is a dimensionless value and is calculated from the impedance of the well with cells and the background impedance of the well without cells.

Briefly, the main steps of the experiment are the following: (i) the background measurement, in which 80 µL complete cell culture medium was loaded to each well of an E-plate in order to establish a constant CI value; (ii) the cell monitoring step, in which 100 µL of cell suspension (10^5^ cell/mL) was added to each well followed by an incubation period of 24 h; and (iii) the compound activity monitoring step, in which test-conjugates (final concentration range: 32 nM–100 µM) were added to the cells at the plateau phase and the CI change was recorded for a further 72 h at 10 kHz. Stock solutions (5 × 10^−3^ M) were prepared from the test-conjugates in distilled water and then further diluted to 320 nM–1 mM in the complete cell culture medium. The wells treated with an adequate volume of pure cell culture medium represented the control. Triplicate wells were used for each treatment for statistical analysis.

Delta CI values—a difference of CI value at the time point of cell inoculation and CI value at a given time point—were calculated by RTCA 2.0 software (ACEA Biosciences, San Diego, CA, USA) integrated into the xCELLigence SP System to characterize the time-dependent antiproliferative/cytotoxic effect of the different GnRH conjugates. IC_50_ values were used to compare the effect of the conjugates. For the calculation of IC_50_ values, Delta CI values obtained at 24, 48 and 72 h for each concentration were normalized to those of the control; then, a sigmoidal dose–response curve was fitted to these data with the nonlinear regression function of OriginPro 8 (OriginLab Corporation, Northampton, MA, USA)

### 4.6. Flow Cytometric Analysis of Cellular Uptake

The cellular uptake of different GnRH conjugates by HT-29 model cells was studied by flow cytometer after a 6 h treatment. The measurement of cellular uptake proceeded very much in the same way as indicated in our previous work [22]. First, 2 × 10^5^ cells/well HT-29 cells were seeded in 12-well plates. After 24 h culturing, the cells were treated with the GnRH conjugates for 6 h. There was a washing step with PBS (phosphate buffer saline, pH = 7.4) to remove the conjugates, and trypsin/EDTA solution (Sigma-Aldrich, St. Louis, MO, USA) was added to dislodge the cells from the plate surface. In the next steps, the cells were centrifuged, washed and resuspended in PBS. During the flow cytometric measurement, at least 10,000 cells were collected, and the intracellular fluorescence intensity of the cells was measured by FACSCalibur flow cytometer (Becton Dickinson, San Jose, CA, USA). Cells treated with complete cell culture medium served as the negative control. Experiments were carried out twice with two parallels per treatment group. The relative fluorescence intensity of Dau built in the conjugates was expressed as geometric mean channel (GeoMean) values and analyzed by CellQuest Pro program (Becton-Dickinson, San Jose, CA, USA) and Flowing 2.5.1 software (Turku Centre of Biotechnology, Turku, Finland). The autofluorescence of the negative control was subtracted from the fluorescence intensity (GeoMean) of the treated samples.

### 4.7. Apoptosis Assay

For determination of the apoptotic effect of GnRH conjugates, annexin V-staining followed by flow cytometry was applied. The steps of apoptosis measurement were basically the same as that described in our previous paper with some minor changes [22]. At first, the cells were inoculated and cultured for 24 h in a 12-well plate; then, they were treated with the free Dau and the conjugates in 100 µM concentration for 6 h and 24 h. During the sample preparation, the cells were removed, washed, resuspended in annexin binding buffer (Becton Dickinson, San Jose, CA, USA) and labelled by annexin V-FITC for 15 min in the dark (Sony Biotechnology, Weybridge, UK). The flow measurement with FACSCalibur flow cytometer (Becton Dickinson, San Jose, CA, USA) was conducted before and after the labelling step by collecting 10,000 cells per each sample in order to eliminate the fluorescence of the cells treated with different Dau-containing conjugate. The apoptosis assay was repeated twice by measuring two parallels.

The data analysis was done with CellQuest Pro (Becton Dickinson, San Jose, CA, USA) and Flowing 2.5.1 (Turku Centre of Biotechnology, Turku, Finland) software. GeoMean values representing the fluorescence intensity of the labelled samples were corrected with that of the non-labelled ones, and the ratio of the viable, annexin V-FITC-positive apoptotic cells in the treated and the control group is displayed in Figure 2.

### 4.8. Molecular Biological Analysis of Apoptosis-Related Genes

#### 4.8.1. RNA Isolation and cDNA Synthesis

A cell viability measurement with the impedance-based xCELLigence SP System was repeated to pinpoint the time points and collect RNA samples for molecular biological studies of apoptosis induced GnRH conjugates. Total cellular RNA was isolated directly from the wells of E-plate with RealTime ready Cell Lysis Kit (Roche Applied Science, Mannheim, Germany) after 24, 48 and 72 h incubation with the conjugates. Cells were washed with ice-cold PBS (phosphate buffer saline, pH = 7.4) before they were lysed with the cell lysis reagent according to the manufacturer’s instruction. RNA samples collected from three parallel wells per treatment group and time point were pooled and stored at −70 °C for further processing.

For cDNA synthesis, the Transcriptor First Strand cDNA Synthesis Kit (Roche Applied Science, Mannheim, Germany) was used by following the manufacturer’s protocol. The synthesis of the cDNA was done right before the qRT-PCR.

#### 4.8.2. Human Apoptosis Gene PCR Array and qRT-PCR

The expression of different apoptosis-related genes was evaluated by a RealTime ready custom panel containing pre-pleated qPCR assays. Our apoptosis custom panels assessed 23 genes involved in different apoptosis pathways (Table 3, Figure 5) three reference genes, and five positive and negative controls in triplicate in a 96-well plate format. The RealTime ready Configurator (https://configurator.realtimeready.roche.com/assaysupply_cp/login.jsf, Roche Applied Science, Mannheim, Germany), a free web-based tool, was applied to design the custom panel and select the target and reference genes with the appropriate primers. The primers used in qRT-PCR were pre-validated and are listed in Table 3. The PCR mix (LightCycler^®^ 480 Probes Master, Roche Applied Science, Mannheim, Germany) and the cDNA samples were added in 20 µL total volume to the wells of the apoptosis custom panel pre-loaded with the different primers. qRT-PCR was performed using the LightCycler^®^ 480 Instrument (Roche Applied Science, Mannheim, Germany) for the quantification of the expression levels of apoptosis-associated genes caused by GnRH conjugates. The PCR program contained a pre-incubation step at 95 °C for 10 min for the activation of FastStart Taq DNA Polymerase and denaturation of the template cDNA; an amplification step, with 55 cycles of 95 °C for 10 s, 60 °C for 30 s and 72 °C for 1 sec; and finally, a cooling step at 40 °C for 30 s. Two representative mRNA samples from each treatment group were analyzed with this panel.

The fold changes in gene expression were calculated using the ΔΔ*C*t method; *GAPDH* and *ACTB* were chosen as housekeeping genes for relative quantification. In the case of conjugates, the results of the relative quantification were normalized to the control treated with the pure medium. A minimum of a two-fold change in the normalized gene expression was regarded as significant. The in-built software of the LightCycler^®^ 480 Instrument and OriginPro 2016 were used for the analysis of gene expression data.

### 4.9. Statistical Analysis

OriginPro 2016 (OriginLab Corporation, Northampton, MA, USA) and MS Excel were used for the statistical analysis of the results. Data obtained from each experiment were expressed as the mathematical means of two or three parallels (depending on the experiment) ± SD. To assess the significance and calculate *p*-values, a one-way ANOVA was used for the data of the cell viability assay, while the Kolmogorov–Smirnov test was performed on the flow cytometric histograms. To compare the difference for all means, a Fisher’s post hoc test was performed. Differences compared to the control were considered statistically different at *p* < 0.05.

## Figures and Tables

**Figure 1 ijms-20-04421-f001:**
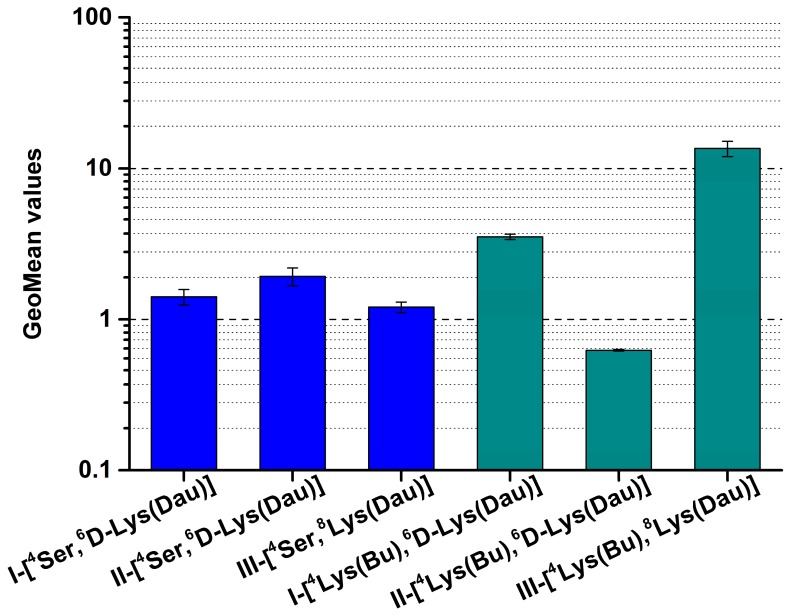
Cellular uptake of GnRH conjugates containing ^4^Ser or ^4^Lys(Bu) by HT-29 cells. Cellular uptake was studied at 10^−4^ M concentration and after 6 h of incubation. The dimensionless GeoMean (geometric mean channel) value refers to the relative fluorescence intensity. Two independent experiments were carried out by using two parallels, and representative data are shown. Data shown represent the mean ± SD of two parallels.

**Figure 2 ijms-20-04421-f002:**
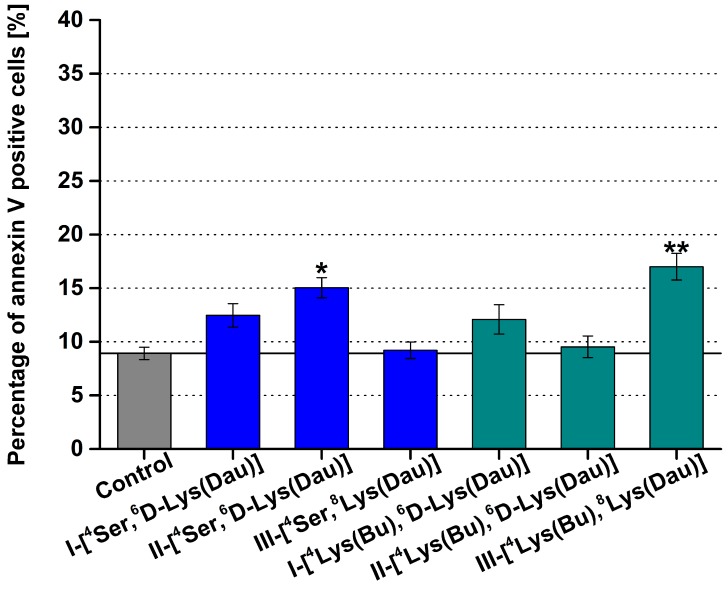
Results of the flow cytometric study of the apoptosis induced by the GnRH conjugates with ^4^Ser or ^4^Lys(Bu). For the treatment, the conjugates were applied at 10^−4^ M concentration for 24 h. Only the viable cells were taken into consideration to determine the percentage of annexin V-positive cells. Two independent experiments were carried out by using two parallels, and representative data are shown. Data shown are mean of two parallels ± SD. The significance levels are the following: *: *p* < 0.05, **: *p* < 0.01.

**Figure 3 ijms-20-04421-f003:**
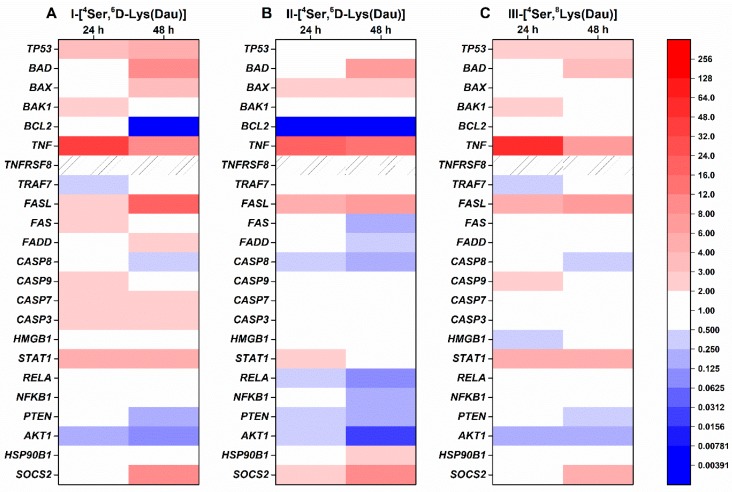
Expression of human apoptosis-related genes in HT-29 cells treated with GnRH-[^4^Ser] conjugates for 24 and 48 h. Effects of I-[^4^Ser,^6^D-Lys(Dau)] (**A**), II-[^4^Ser,^6^D-Lys(Dau)] (**B**), III-[^4^Ser,^8^Lys(Dau)] (**C**) on gene expression were analyzed by a human apoptosis gene PCR array (RealTime ready custom panel). The colors of the heatmap show significant fold changes in gene expression compared to control. Fold changes ≥ 2 and *p* < 0.05 were considered as significant. A hashed zone means invalid PCR results. *TP53*: tumor protein p53 data; *BAD*: BCL2-associated agonist of cell death; *BAX*: BCL2-associated X protein; *BAK1*: BCL2-antagonist/killer 1; *BCL2*: B-cell CLL/lymphoma 2; *TNF*: tumor necrosis factor (TNF)-alpha, tumor necrosis factor ligand superfamily member 2; *TNFRSF8*: tumor necrosis factor receptor superfamily, member 8; *TRAF7*: TNF receptor-associated factor 7; *FASL*: Fas ligand, TNF superfamily member 6 (TNFSF6); *FAS*: TNF receptor superfamily member 6 (TNFRSF6); *FADD*: Fas (TNFRSF6)-associated via death domain; *CASP7*: caspase 7; *CASP3*: caspase 3; *CASP9*: caspase 9; *CASP8*: caspase 8; *HMGB1*: high-mobility group box 1; *NFKB1*: nuclear factor of kappa light polypeptide gene enhancer in B-cells 1; *RELA*: v-rel reticuloendotheliosis viral oncogene homolog A; *AKT1*: v-akt murine thymoma viral oncogene homolog 1; *PTEN*: phosphatase and tensin homolog; *STAT1*: signal transducer and activator of transcription 1; *SOCS2*: suppressor of cytokine signaling 2; *HSP90B1*: heat shock protein 90 kDa beta (Grp94) member 1.

**Figure 4 ijms-20-04421-f004:**
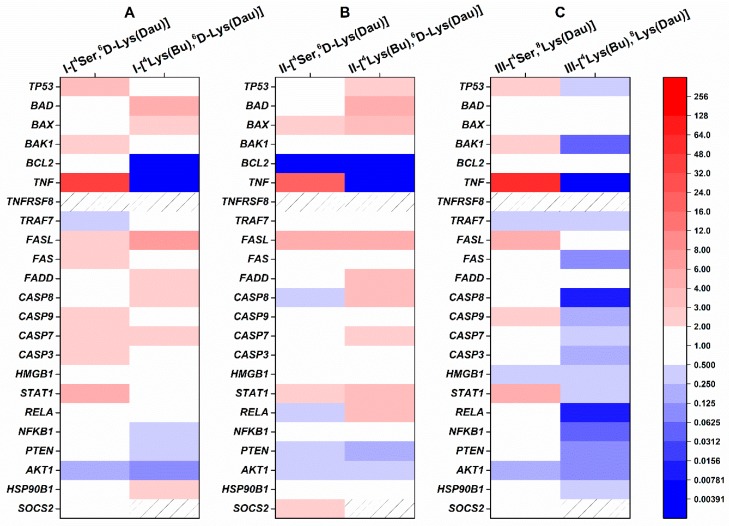
Comparison of the expression of human apoptosis-related genes in HT-29 cells treated with GnRH-[^4^Ser] and GnRH-[^4^Lys(Bu)] conjugates for 24 h. Effects of the conjugates pairs of GnRH-I (**A**), GnRH-II (**B**), GnRH-III (**C**) on gene expression were analyzed by a human apoptosis gene PCR array (RealTime ready custom panel). The colors of the heatmap show significant fold changes in gene expression compared to control. Fold changes ≥ 2 and *p* < 0.05 were considered as significant. The hashed zone means invalid PCR results. *TP53*: tumor protein p53 data; *BAD*: BCL2-associated agonist of cell death; *BAX*: BCL2-associated X protein; *BAK1*: BCL2-antagonist/killer 1; *BCL2*: B-cell CLL/lymphoma 2; *TNF*: TNF-alpha, Tumor necrosis factor ligand superfamily member 2; *TNFRSF8*: Tumor necrosis factor receptor superfamily, member 8; *TRAF7*: TNF receptor-associated factor 7; *FASL*: Fas ligand, TNF superfamily member 6 (TNFSF6); *FAS*: TNF receptor superfamily member 6 (TNFRSF6); *FADD*: Fas (TNFRSF6)-associated via death domain; *CASP7*: caspase 7; *CASP3*: caspase 3; *CASP9*: caspase 9; *CASP8*: caspase 8; *HMGB1*: high-mobility group box 1; *NFKB1*: nuclear factor of kappa light polypeptide gene enhancer in B-cells 1; *RELA*: v-rel reticuloendotheliosis viral oncogene homolog A; *AKT1*: v-akt murine thymoma viral oncogene homolog 1; *PTEN*: phosphatase and tensin homolog; *STAT1*: signal transducer and activator of transcription 1; *SOCS2*: suppressor of cytokine signaling 2; *HSP90B1*: heat shock protein 90 kDa beta (Grp94) member 1.

**Figure 5 ijms-20-04421-f005:**
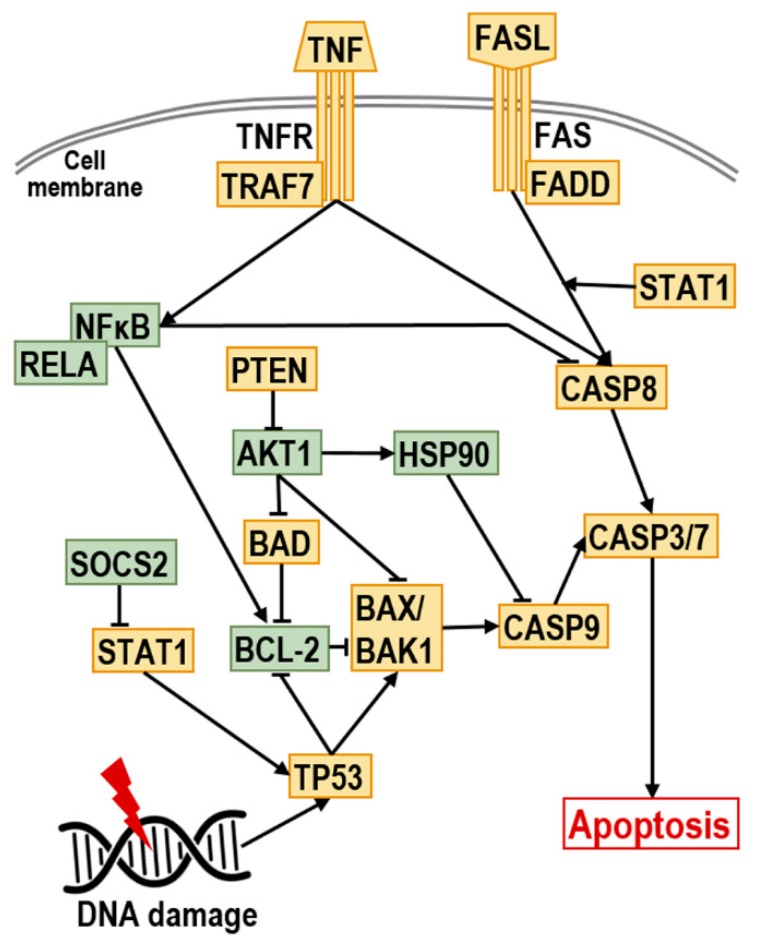
Pathway of the investigated genes involved in apoptosis. TP53: tumor protein p53 data; BAD: BCL2-associated agonist of cell death; BAX: BCL2-associated X protein; BAK1: BCL2-antagonist/killer 1; BCL2: B-cell CLL/lymphoma 2; TNF: TNF-alpha, Tumor necrosis factor ligand superfamily member 2; TNFRSF8: Tumor necrosis factor receptor superfamily, member 8; TRAF7: TNF receptor-associated factor 7; FASL: Fas ligand, TNF superfamily member 6 (TNFSF6); FAS: TNF receptor superfamily member 6 (TNFRSF6); FADD: Fas (TNFRSF6)-associated via death domain; CASP7: caspase 7; CASP3: caspase 3; CASP9: caspase 9; CASP8: caspase 8; HMGB1: high-mobility group box 1; NFKB1: nuclear factor of kappa light polypeptide gene enhancer in B-cells 1; RELA: v-rel reticuloendotheliosis viral oncogene homolog A; AKT1: v-akt murine thymoma viral oncogene homolog 1; PTEN: phosphatase and tensin homolog; STAT1: signal transducer and activator of transcription 1; SOCS2: suppressor of cytokine signaling 2; HSP90B1: heat shock protein 90 kDa beta (Grp94) member 1.

**Table 1 ijms-20-04421-t001:** Daunorubicin–gonadotropin-releasing hormone (Dau–GnRH) conjugates investigated in this study.

Conjugate	Code
GnRH-I-[^4^Ser,^6^D-Lys(Dau=Aoa)]	I-[^4^Ser,^6^D-Lys(Dau)]
GnRH-II-[^4^Ser,^6^D-Lys(Dau=Aoa)]	II-[^4^Ser,^6^D-Lys(Dau)]
GnRH-III-[^4^Ser,^8^Lys(Dau=Aoa)]	III-[^4^Ser,^8^Lys(Dau)]
GnRH-I-[^4^Lys(Bu),^6^D-Lys(Dau=Aoa)]	I-[^4^Lys(Bu),^6^D-Lys(Dau)]
GnRH-II-[^4^Lys(Bu),^6^D-Lys(Dau=Aoa)]	II-[^4^Lys(Bu),^6^D-Lys(Dau)]
GnRH-III-[^4^Lys(Bu),^8^Lys(Dau=Aoa)]	III-[^4^Lys(Bu),^8^Lys(Dau)]

**Table 2 ijms-20-04421-t002:** IC_50_ values of cytotoxicity of Dau–GnRH–[^4^Ser/^4^Lys(Bu)] conjugates determined on HT-29 cell line.

Conjugates		IC_50_ ^1^ (µM)	
	24 h	48 h	72 h
I-[^4^Ser,^6^D-Lys(Dau)]	>100	60.25 ± 3.50	21.94 ± 1.54
II-[^4^Ser,^6^D-Lys(Dau)]	58.88 ± 10.04	24.20 ± 0.86	19.73 ± 2.51
III-[^4^Ser,^8^Lys(Dau)]	>100	>100	56.82 ± 5.44
I-[^4^Lys(Bu),^6^D-Lys(Dau)]	>100	18.71 ±1.19	16.18 ± 1.75
II-[^4^Lys(Bu),^6^D-Lys(Dau)]	65.78 ± 3.89	69.70 ± 4.12	48.08 ± 6.89
III-[^4^Lys(Bu),^8^Lys(Dau)]	10.25 ± 1.66	4.26 ± 0.89	4.56 ± 0.27

^1^ IC_50_ values represent the mean ± SD of three parallel measurements and were calculated by fitting a sigmoidal dose–response curve with OriginPro 2016 software.

**Table 3 ijms-20-04421-t003:** The apoptosis-related genes investigated and the primers used in this study.

	Gene	Gene Name	Forward and Reverse Primers
**Intrinsic Apoptotic Pathway**	*TP53*	Tumor protein p53 data	AGGCCTTGGAACTCAAGGAT, CCCTTTTTGGACTTCAGGTG
	*BAD*	BCL2-associated agonist of cell death	CTACGGTGGGAGAGGAAGC, TGTTACGTAGTCAAGGCACAGC
	*BAX*	BCL2-associated X protein	GGACGTGGGCATTTTTCTTA, GTTTATTACCCCCTCAAGACCA
	*BAK1*	BCL2-antagonist/killer 1	AGACCTGAAAAATGGCTTCG, CGGAAAACCTCCTCTGTGTC
	*BCL2*	B-cell CLL/lymphoma 2	GCACCTGCACACCTGGAT, AGCCAGGAGAAATCAAACAGAG
**Extrinsic Apoptotic Pathway**	*TNF*	TNF-alpha, tumor necrosis factor ligand superfamily member 2	CGGTGCTTGTTCCTCAGC, GCCAGAGGGCTGATTAGAGA
	*TNFRSF8*	Tumor necrosis factor receptor superfamily, member 8	GCTGTCAGGAGGTGCTGTTAC, GTAGGCCTCTGTGGGCACT
	*TRAF7*	TNF receptor-associated factor 7	ATGTCTCTGCGCTCCACATT, AGCTGACAGCACAGCTTCAC
	*FASL*	Fas ligand, TNF superfamily, member 6	CAGTTCTTCCCTGTCCAACC, GTGGTGGTGGCCTCCTTT
	*FAS*	TNF receptor superfamily, member 6	TGAGGAAGACTGTTACTACAGTTGAGA, GCAGTCCCTAGCTTTCCTTTC
	*FADD*	Fas (TNFRSF6)-associated via death domain	AGGTAGCCCAGCACTGTGAA, AGGTGGTCTGTGGCTCACTC
**Effector Apoptotic Proteins**	*CASP7*	Caspase 7	CCCGCAAAGCAACGTCTA, CCCCTGCTCTTCAATACAGC
	*CASP3*	Caspase 3	CTGGTTTTCGGTGGGTGT, CAGTGTTCTCCATGGATACCTTTATT
	*CASP9*	Caspase 9	CCCAAGCTCTTTTTCATCCA, AGTGGAGGCCACCTCAAAC
	*CASP8*	Caspase 8	GAAAGGGTGGAGCGGATT, GATTTCTGCTGAAGTCCATCTTTT
	*HMGB1*	High-mobility group box 1	GAGTGAGGAGGCTGCGTCT, TGCCCATGTTTAGTTATTTTTCC
**Cytokine/Growth-Factor Signaling Pathway**	*NFKB1*	Nuclear factor of kappa light polypeptide gene enhancer in B-cells 1	CTGGCAGCTCTTCTCAAAGC, TCCAGGTCATAGAGAGGCTCA
	*RELA*	v-rel reticuloendotheliosis viral oncogene homolog A	ACCGCTGCATCCACAGTT, GGATGCGCTGACTGATAGC
	*AKT1*	v-akt murine thymoma viral oncogene homolog 1	GCAGCACGTGTACGAGAAGA, GGTGTCAGTCTCCGACGTG
	*PTEN*	Phosphatase and tensin homolog	GCTACCTGTTAAAGAATCATCTGGA, CTGGCAGACCACAAACTGAG
	*STAT1*	Signal transducer and activator of transcription 1	GGATTGAAAGCATCCTAGAACTCA, GATGAAGCCCATGATGCAC
**Anti-Apoptotic Proteins**	*SOCS2*	Suppressor of cytokine signaling 2	AGGCCTCACTGCAATTTGAT, TGCAAAATATAAAATGCCCAAG
	*HSP90B1*	Heat shock protein 90 kDa beta (Grp94), member 1	CTGGAAATGAGGAACTAACAGTCA, TCTTCTCTGGTCATTCCTACACC

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
