# Peer review of "Apoptotic Effects of Drug Targeting Conjugates Containing Different GnRH Analogs on Colon Carcinoma Cells"

_ijms, 2019, doi:10.3390/ijms20184421_

Round 1
Reviewer 1 Report
Major comments
In this paper, authors compared the cytotoxic/apoptotic activity of mutiple forms of GnRH-based conjugates in a human colon carcinoma cells. Experiments were well-desingned, the methods are reasonable and the narrative flows well. Therefore, only minor comments listed below should be addressed.
minor comments are listed below:
Line 77: “result” should be “results”.
Line 270: “FASLG” should be “FASL”. Though the manuscript, FASL is used in the text part although FASLG is found in Figures and Figure legends. It should be consistent.
Line 307: “24” should be “24 h”.
Lines 351-352: “The fact that …..the targeted therapy” seems to be grammatically wrong.
Line 410: “are” should be inserted between “studies” and “available”.
Line 471: “found” should be “find”.
Line 501: “DIEA” should be “DIPEA”.
There are possibly other careless and grammatical mistakes through the manuscript. Please check it out, again.
Author Response
We deeply appreciate the Reviewers’ comments, corrections and most importantly the time they have spent on judging our manuscript. We have made all efforts to address all concerns that have been raised. Below you can find a point-by-point listing of our replies. The Reviewers’ comments are written in blue, our response is written in black and itself the text with the changes is also shown in Italics and in inverted commas where all changes that we made in the manuscript are described in red color.
As practically all of the suggested modifications of Reviewers were accepted and we also paid attention to answering the suggestions mentioned, we find that the revised version of the manuscript is significantly improved and Authors are grateful for the Editor and the Reviewers’ help in getting this done.
Response to Referee 1
Moderate English changes required.
…careless and grammatical mistakes through the manuscript. Please check it out, again.
Authors apologize for the grammar mistakes and they tried to do their best to minimize the grammatical inaccuracies. A native speaker was also asked to carefully check and correct the spelling and grammatical errors.
“…but results in a further decrease of…”
“Figure 1. Comparison of the expression of human apoptosis-related genes in HT-29 cells treated with GnRH-[4Ser] and GnRH-[4Lys(Bu)] conjugates for 24 h.”
“The fact that GnRH analogs are widely known for possessing their antitumor activity, which represents a valuable benefit for their use in the targeted therapy [1].”
“Only couple of studies are available in the literature …”
“By detecting the expression of 23 apoptosis-related genes we could found find further evidence for…”
“…after neutralization (10% DIPEA in DCM)…”
“Acknowledgements: … Authors express their gratitude to Dr A. Tziakouri for proofreading and technical language support”
Though the manuscript, FASL is used in the text part although FASLG is found in Figures and Figure legends. It should be consistent.
Authors apologize for this inconsistency. For the abbreviation of Fas Ligand, FASL is used uniformly in the text part, Figures and Figure legends of the manuscript as well as in the supporting material.
Reviewer 2 Report
Lajko et al provide an interesting set of experiments that characterize te effect of several GnRH conjugates on cell viability in a colon cancer model. the analysis is quite thorough however few aspects should be improved:
1) it is important that also efficacy of the conjugates is reported and not only their potency (i.e., table 1)
2) the paper would be stronger if other ways to assess cell viability are used - for example cytotoxicity assay, cPARP1, senescence assay
3) Figure 1 and 2 should have some time-course analysis done. also, the lack of error bars (need more replicates) is problematic to assess significance. also, calculating p value off two replicates is not appropriate
4) it is unclear how many times the qPCR panels have been repeated
Author Response
We deeply appreciate the Reviewers’ comments, corrections and most importantly the time they have spent on judging our manuscript. We have made all efforts to address all concerns that have been raised. Below you can find a point-by-point listing of our replies. The Reviewers’ comments are written in blue, our response is written in black and itself the text with the changes is also shown in Italics and in inverted commas where all changes that we made in the manuscript are described in red color.
As practically all of the suggested modifications of Reviewers were accepted and we also paid attention to answering the suggestions mentioned, we find that the revised version of the manuscript is significantly improved and Authors are grateful for the Editor and the Reviewers’ help in getting this done.
Response to Reviewer 2
It is important that also efficacy of the conjugates is reported and not the potency (i.e. table 1)
Authors agree that both potency (concentration/dose (i.e. IC50) of a drug that is needed to elicit a given effect) and efficacy (maximum effect of a drug) are important parameters to evaluate and compare the activity of different therapeutics. Nevertheless, in the scientific literature, it is well-accepted that only the potency – IC50 values – are usually published to characterize the antitumor activity of different chemotherapeutic agents including drug-targeting conjugates.
The real-time data provided by the impedance-based xCELLigence system could be used to analyze and compare the efficacy of the GnRH conjugates. The real-time cell index curves were originally presented in the supplementary material. For the better comparison of the maximum antitumor effect of the conjugates, we could calculate the viability (ratio of viable cells expressed as a percentage of control) of HT-29 cells treated with the different conjugates at the highest concentration (10-4 M) by using the real-time cell indices. The supplementary material was completed with Table S1 containing these viability results and the manuscript was modified and clarified according to these data.
“The IC50 values showing the potency (Table 2) and cell viability (viab.) percentages showing the efficacy of the conjugates (Table S1) were calculated from the results of the time-course study obtained after 24, 48 and 72 h incubation time.
In case of the conjugates built on native GnRH conjugates, the II-[4Ser,6D-Lys(Dau)] proved to be the most effective potent one followed by the I-[4Ser,6D-Lys(Dau)] and III-[4Ser, 8Lys(Dau)] (Table 1). However In long-term (after 72 h incubation), there was no a significant difference (p < 0.014) in the long-term cytotoxic effect efficacy of the GnRH-I and GnRH-II conjugates. However, I-[4Ser,6D-Lys(Dau)] showed the strongest antitumor effect (viab.: 4,75%, Table S1) at 10-4 M concentration, but according to the time-course study, the II-[4Ser,6D-Lys(Dau)] elicited a more immediate cytotoxic effect (viab24h: 37.18% vs viab24h for I-[4Ser,6D-Lys(Dau)]: 150.05%; Figure S1, Table S1). III-[4Ser, 8Lys(Dau)] had showed about threefold weaker IC50 value (Table 1) and the maximal tumor growth inhibitory effect was manifested only at 10-4 M concentration (viab72h: 19.05%) and only after 72 h incubation (Table 12).
Table 1 Cytotoxic effect IC50 values of cytotoxicity of Dau-GnRH-[4Ser/4Lys(Bu)] conjugates determined on HT-29 cell line”
“The substitution with 4Lys(Bu) proved to modify the cytotoxic effect of the conjugates depending on the type of GnRH analog. The most significant change was detected in case of the GnRH-III based conjugates. As it was expected, the replacement of 4Ser by 4Lys(Bu) led to a more than one order of magnitude smaller IC50 value (Table 1) and 10 fold a stronger antitumor activity with earlier onset (Table S1 Table 2). In the case of GnRH-I conjugates, this type of modifications could cause only a slight increase in the cytotoxic effect potency (smaller IC50 value) after 72 h, but the onset of the cytotoxic activity took less time (48 h for I-[4Lys(Bu),6D-Lys(Dau)] vs. 72 h for I-[4Ser,6D-Lys(Dau)]. On the contrary, IC50 values of GnRH-II conjugate with 4Lys(Bu) (II-[4Lys(Bu),6D-Lys(Dau)]) was more than two times higher than that of II-[4Ser,6D-Lys(Dau)] after 48 and 72 h incubation (Table 1).”
“Table S1 Efficacy of the Dau-GnRH-[4Ser/4Lys(Bu)] conjugates – Viability of HT-29 cells treated with the Dau-GnRH-[4Ser/4Lys(Bu)] conjugates at 10-4 M for 24, 48 and 72 h
|
Conjugates |
Viability 1 (%) (control = 100 % ± 5.97-6,52) |
||
|
24 h |
48 h |
72 h |
|
|
I-[4Ser,6D-Lys(Dau)] |
150.05 ± 8.55 |
43.82 ± 1.18 |
4.75 ± 1.59 |
|
II-[4Ser,6D-Lys(Dau)] |
37.18 ± 3.08 |
16.91 ±1.91 |
20.43 ± 2.66 |
|
III-[4Ser ,8Lys(Dau)] |
10.42 ± 11.48 |
68.63 ± 5.13 |
19.05 ± 0.97 |
|
I-[4Lys(Bu),6D-Lys(Dau)] |
86.69 ± 4.84 |
15.50 ± 0.55 |
5.31 ± 1.27 |
|
II-[4Lys(Bu),6D-Lys(Dau)] |
35.64 ± 3.41 |
19.40 ± 2.77 |
24.31 ± 4.20 |
|
III-[4Lys(Bu),8Lys(Dau)] |
57.8 ± 0.80 |
12.52 ± 0.50 |
8.77 ± 0.61 |
1 Each data represents the mathematical average of three parallels ± SD. The decrease in the Cell indices caused by the different treatments was normalized to the identical control and this value was given as ‘Viability’ in percent.”
The paper would be stronger if other ways to assess cell viability are use – for example cytotoxicity assay, cPARP1, senescence assay.
Authors understand that the impedance-based xCELLigence system is not a classical cell viability assay. However, the increasing number of publications have already proved that impedimetry is sensitive enough to distinguish the viable and dead cells. In contrast to the above-mentioned technics, which are endpoint assays, this system detects the impedance changes in a real-time manner, which provides further information about e.g. rapid morphological changes induced by a given drug, mechanism of drug action, development of drug resistance and etc. Previous studies have demonstrated a good correlation between impedance measurements and endpoint viability assays. It was also mentioned in the manuscript, that “In the case of GnRH-III conjugates, the beneficial effect of this modification with 4Lys(Bu) was well-established in our previous studies [21, 43]. Thus, this pair of GnRH-III conjugates was considered as reference conjugates in our present study.” In the aforementioned papers MTT- or alamarBlue-assay was used to determine the cytotoxic effects of the GnRH-III conjugates, and a similar difference was found between the GnRH-III conjugates with 4Ser and 4Lys(Bu). These findings clearly indicate that the xCELLigence SP System is a proper and suitable method to assess cell viability.
Figure 1 and 2 should have some time-course analysis done
The incubation times applied during the cellular uptake and apoptotic measurements were chosen on the basis of previous studies and preliminary experiments. The relatively long incubation time (6 h) in the cellular uptake measurement can be also explained by the fact that the mammalian type I GnRH receptor lacks an intracellular cytoplasmic C-terminal tail, which could imply a relatively slow receptor internalization. In case of the apoptosis assay, the 24 h of incubation time was proved to be the shortest time period, which could already evoke slight cytotoxic effect, but the adequate ratio of viable cells was also present in the sample to perform an Annexin V staining.
Authors agree with Reviewer 2 that our present research has raised some questions in need of further investigations (time-course analysis of the apoptotic effect and cellular uptake or cell viability study in a wider level). However, future works on these topics exceed the time limits of the submission of this paper to the special issue entitled The Role of Gonadotropin-Releasing Hormone Receptor in Human Diseases.
Also, the lack of error bars (need more replicates) is problematic to assess significance. also, calculating p value off two replicates is not appropriate.
The Authors apologize for missing the indication of the standard deviation in Figure 1 and 2. Cellular uptake and apoptosis experiments were repeated twice by measuring 2 parallels. Since similar results were obtained in the two independent measurements, we decided to use a representative set of data for the statistical analysis. Histograms obtained from FACSCalibur cytometer were further analyzed by Kolmogorov–Smirnov test, which is generally accepted, valid test for comparing flow cytometric histograms, to assess the significance and calculate the p-value. Therefore, these figures have been completed with the error bars as well as the Methods section and Figure legends were also enriched with the description of data depicted in the graphs.
“Experiments were carried out twice with two parallels per treatment groups. The relative fluorescence …”
“Apoptosis assay was repeated twice by measuring two parallels. The data analysis was done with CellQuest Pro”
“Data obtained from each experiment were expressed as the mathematical means of two or three parallels (depending on the experiment) ±SD. To assess the significance and calculate p values analyzed by one-way ANOVA was used for the data of cell viability assay, while Kolmogorov–Smirnov test was done on the flow cytometric histograms.”
“Figure 2. Cellular uptake of GnRH conjugates containing 4Ser or 4Lys(Bu) by HT-29 cells.
Cellular uptake was studied at 10-4 M concentration and after 6 h incubation. The dimensionless GeoMean (geometric mean channel) value refers to the relative fluorescence intensity. Two independent eExperiments were carried out in duplicates by using two parallels and representative data are shown. Data shown represent the mean ± SD of two parallels.”
“Figure 3. Results of the flow cytometric study of the apoptosis induced by the GnRH conjugates with 4Ser or 4Lys(Bu)
For the treatment, the conjugates were applied at 10-4 M concentration for 24 h. Only the viable cells were taken into consideration to determine the ‘percentage of Annexin V positive cells. Two independent eExperiments were carried out in duplicates by using two parallels and representative data are shown. Data shown are mean of two parallels ± SD. The significance levels are the followings: *: p < 0.05, **: p < 0.01.”
It is unclear how many times the qPCR panels have been repeated.
It escaped the Authors’ attention that the information about the replicates was missing from the description of molecular biological analysis. Our RealTime ready Custom panel contains 32 pre-plated qPCR assays for 23 target genes, 3 reference genes and 5 positive and negative controls in triplicates. Three blocks of these genes can be found in a 96-well plate. Two representative mRNA samples from each treatment group were analyzed with this panel. Most of the aforementioned information was inserted to the manuscript (4.8.2. Human apoptosis gene PCR array and qRT-PCR)
“Our apoptosis custom panels assessed 23 genes involved in different apoptosis pathways (Table 3, Figure 5) 3 reference genes and 5 positive and negative controls in triplicates in a 96-well plate format.”
“…and finally cooling step at 40°C for 30 sec. Two representative mRNA samples from each treatment group were analyzed with this panel.”